# Pre- and Neonatal Exposure to Lead (Pb) Induces Neuroinflammation in the Forebrain Cortex, Hippocampus and Cerebellum of Rat Pups

**DOI:** 10.3390/ijms21031083

**Published:** 2020-02-06

**Authors:** Karina Chibowska, Jan Korbecki, Izabela Gutowska, Emilia Metryka, Maciej Tarnowski, Marta Goschorska, Katarzyna Barczak, Dariusz Chlubek, Irena Baranowska-Bosiacka

**Affiliations:** 1Department of Biochemistry and Medical Chemistry, Pomeranian Medical University, Powstańców Wlkp. 72 St., Szczecin, 70-111 Szczecin, Poland; zara1988@wp.pl (K.C.); jan.korbecki@onet.eu (J.K.); emilia_metryka@o2.pl (E.M.); rcmarta@wp.pl (M.G.); dchlubek@pum.edu.pl (D.C.); 2Department of Medical Chemistry, Pomeranian Medical University, Powstańców Wlkp. 72 St., 70-111 Szczecin, Poland; izagut@pum.edu.pl; 3Department of Physiology, Pomeranian Medical University, Powstańców Wlkp. 72 St., 70-111 Szczecin, Poland; maciejt@sci.pum.edu.pl; 4Department of Conservative Dentistry and Endodontics, Pomeranian Medical University, Powstańców Wlkp. 72 St., 70-111 Szczecin, Poland; kasiabarczak@vp.pl

**Keywords:** lead (Pb), neuroinflammation, cytokines, transforming growth factor-β, interleukin 1β, cyclooxygenases, prostaglandin E_2_

## Abstract

Lead (Pb) is a heavy metal with a proven neurotoxic effect. Exposure is particularly dangerous to the developing brain in the pre- and neonatal periods. One postulated mechanism of its neurotoxicity is induction of inflammation. This study analyzed the effect of exposure of rat pups to Pb during periods of brain development on the concentrations of selected cytokines and prostanoids in the forebrain cortex, hippocampus and cerebellum. Methods: Administration of 0.1% lead acetate (PbAc) in drinking water ad libitum, from the first day of gestation to postnatal day 21, resulted in blood Pb in rat pups reaching levels below the threshold considered safe for humans by the Centers for Disease Control and Prevention (10 µg/dL). Enzyme-linked immunosorbent assay (ELISA) method was used to determine the levels of interleukins IL-1β, IL-6, transforming growth factor-β (TGF-β), prostaglandin E2 (PGE2) and thromboxane B2 (TXB2). Western blot and quantitative real-time PCR were used to determine the expression levels of cyclooxygenases COX-1 and COX-2. Finally, Western blot was used to determine the level of nuclear factor kappa B (NF-κB). Results: In all studied brain structures (forebrain cortex, hippocampus and cerebellum), the administration of Pb caused a significant increase in all studied cytokines and prostanoids (IL-1β, IL-6, TGF-β, PGE2 and TXB2). The protein and mRNA expression of COX-1 and COX-2 increased in all studied brain structures, as did NF-κB expression. Conclusions: Chronic pre- and neonatal exposure to Pb induces neuroinflammation in the forebrain cortex, hippocampus and cerebellum of rat pups.

## 1. Introduction

In recent years, the use of lead (Pb) in industry and other areas of life has been significantly reduced thanks to the introduction of numerous prevention measures [1,2,3,4,5]. Despite these positive changes, Pb remains one of the substances representing the greatest potential risk to human health. Ranked the second most toxic substance on the priority list created by the Agency for Toxic Substances and Disease Registry (ATSDR) in 2017 [6], Pb poses a particular risk to the nervous system, especially in early life [7,8,9,10,11]. It is linked to numerous metabolic and structural changes in the developing brain (for detailed review see [12,13,14,15]), resulting in various cognitive disorders [16,17,18,19]. Children exposed to Pb in the early stages of development show a high incidence of many neurodevelopmental diseases, such as hyperactivity, emotional symptoms and conduct disorder [20], attention-deficit hyperactivity disorder (ADHD) [21,22], autism spectrum disorder (ASD) [23,24,25,26], and tend to have lower IQs [27,28]. Until recently, the safe blood Pb concentration was believed to be 10 μg/dL [29,30]. However, it is now known even much lower blood Pb levels can cause neurobehavioral deficits and lower IQ scores [10,11,31,32,33,34]. Hence, the currently postulated “threshold level” for children and pregnant women remains at 5 μg/dL [35].

Research on the mechanisms of Pb’s neurotoxic effects has been conducted for over a decade. Also our team has been researching the mechanisms of neurotoxic action of Pb for many years, using an animal model of pre- and neonatal administration of Pb that imitates the environmental exposure. In our previous papers, we demonstrated morphological and histological changes in rat brains, confirmed at a cellular level by electron microscopy and biochemical and molecular studies [9,10]. We also proved that pre- and neonatal exposure to Pb results in numerous disorders, as confirmed by behavioral and cognitive tests [36,37]. It is also known so far that Pb impairs the functioning of progenitor cells, leading to structural changes in the hippocampus [16,38,39] and cerebral cortex [40,41]. Pb also disrupts developmental cortical plasticity, causing dysfunction in neurodevelopment [25]. Moreover, prenatal exposure to Pb affects the activity and expression of key synaptic proteins [10], impairs the functioning of synaptic mitochondria [7,42], disrupts the energy metabolism of neurons and astrocytes, and leads to impaired metabolic cooperation between these cells [43].

Indirect evidence suggests that inflammation may be one of the major mechanisms of Pb neurotoxicity. For example, prenatal neuroinflammation induced by lipopolysaccharide (LPS) in rats and in vitro studies shows effects similar to Pb exposure. LPS-induced neuroinflammation disturbs neurogenesis [44,45], inhibits maturation of oligodendrocyte progenitor cells [46] and inhibits nerve myelination [47,48,49,50]. Uncontrolled chronic neuroinflammation causes neural cell death [51,52,53,54], which combined with the suppressed plasticity in the brain leads to malformations in this organ [55]. For this reason, some researchers implicate neuroinflammation in ASD [56,57].

Although the proinflammatory effect of Pb has been observed in many cells, tissues and organs [58], few studies have focused on the brain [59,60,61,62]. Therefore, the aim of this study was to analyze how exposure to Pb during brain development affects the concentrations of select cytokines and prostanoids in the forebrain cortex, hippocampus and cerebellum of rat pups. Our study was carried out on the model pre- and neonatal administration of Pb, i.e., during fetal life and breastfeeding, which imitates the environmental Pb exposure in low dose, resulting in blood Pb levels below the threshold previously considered “safe” (10 µg/dL).

The study involved analysis of proinflammatory cytokines (interleukin 1β (IL-1β), interleukin 6 (IL-6), and transforming growth factor-β (TGF-β)); prostanoids (prostaglandin E_2_ (PGE_2_) and thromboxane B_2_ (TXB_2_)); and the main enzymes responsible for producing the studied prostanoids: cyclooxygenase 1 (COX-1) and cyclooxygenase 2 (COX-2).

## 2. Results

### 2.1. Lead Concentration in Whole Blood and Brain Tissue

The lead exposure regimen used in our study (from the first day of fetal life and subsequent feeding by the mother to 21 PND) caused a statistically significant increase in the concentration of Pb in whole blood (measured at 28 PND), with final levels reaching 6.50 ± 0.14 μg/dL in the study group compared to 0.07 ± 0.15 μg/dL in the control group (*p* = 0.002).

The Pb concentration was significantly higher in all brain structures tested compared to control. It was highest in the hippocampus in both study and control groups (7.50 ± 0.43 µg/dL and 0.31 ± 0.25 µg/dL, respectively (*p* = 0.001)), followed by the cerebellum (7.52 ± 0.21 µg/dL vs. 0.03 ± 0.02 µg/dL (*p* = 0.001)) and forebrain cortex (7.20 ± 0.12 µg/dL vs. 0.03 ± 0.01 µg/dL (*p* = 0.002)). Brain Pb levels showed a strong positive correlation with whole blood Pb levels (cortex: Rs = +0.68; cerebellum: Rs = +0.65; hippocampus: Rs = +0.75; *p* < 0.005 for all examined brain structures).

### 2.2. Exposure to Lead Increases the Level of Cytokines in the Brain

Exposure of rat pups to Pb resulted in a statistically significant increase in IL-1β, IL-6 and TGF-β in the forebrain cortex, hippocampus and cerebellum (Figure 1). The highest levels of all three cytokines were observed in the hippocampus in both the study and control groups.

IL-1β increased by 90% in the forebrain cortex, by 90% in the cerebellum, and by 40% in the hippocampus. IL-6 increased by 120% in the forebrain cortex, by 110% in the hippocampus, and by 30% in the cerebellum. TGF-β increased most (by 120%) in the cerebellum, followed by the forebrain cortex (by 80%) and the hippocampus (by 70%).

### 2.3. Lead-Induced Neuroinflammation Increases the Synthesis of Prostanoids

In our study Pb not only increased cytokine levels in the brain but also affected the synthesis pathway producing prostaglandins and thromboxanes (Figure 2). This effect was much more pronounced than the increase in cytokines. In the cerebellum Pb increased the concentration of PGE_2_ by 175% relative to control. In turn, in the forebrain cortex and hippocampus the Pb-induced increase in PGE_2_ was 125% and 105% relative to control, respectively. In the forebrain cortex, the concentration of PGE_2_ was the highest.

Similar results were obtained for TXB_2_. The highest increase in this prostanoid was observed in the forebrain cortex. In this structure, the Pb-induced increase in TXB_2_ was 190% compared to control. Similarly to PGE_2_, the forebrain cortex showed the highest concentration of TXB_2_. A smaller effect of Pb on TXB_2_ was observed in the cerebellum and hippocampus (increases by 125% and 95%, respectively).

### 2.4. Increased Levels of Prostanoids in Lead-Induced Neuroinflammation are Accompanied by Increased Expression of Cyclooxygenases.

The Pb-induced increase in the concentrations of PGE_2_ and TXB_2_ was accompanied by an increase in the expression of enzymes involved in their synthesis (Figure 3). COX-1 mRNA expression increased by 75% in the cerebellum, by 60% in the forebrain cortex, and by 60% in the hippocampus, compared to control. Similar results were obtained by analysis of COX-1 protein expression, which increased by around 30% in all three brain structures.

The Pb-induced increase in COX-2 mRNA expression was similar to that of COX-1 (Figure 4). The greatest influence of Pb was seen in the cerebellum where COX-2 mRNA expression increased by 100% compared to control, followed by the hippocampus (by 90%) and the forebrain cortex (by 70%). COX-2 protein expression increased by 140% in the cerebellum, by 50% in the forebrain cortex and by 50% in the hippocampus.

### 2.5. Exposure to Lead Increases NF-κB Concentration in the Brain

Exposure of rat pups to lead caused an increase in NF-κB levels in the brain (Figure 5). The highest increase was observed in the hippocampus compared to the other structures studied (80% higher than control). The hippocampus also showed the highest concentration of all tested cytokines. Lower NF-κB level but also still significant increases were observed in the forebrain cortex and cerebellum (by 50% and 40%, respectively).

## 3. Discussion

Cytokines are small proteins regulating the immune cells, which can be divided according to their production by respective T helper (Th) cells [63]. Hence we can distinguish proinflammatory cytokines Th1 (IFN-γ and TNF-α) and Th17 (IL-17) and anti-inflammatory cytokines Th2 (IL-4, IL-5, IL-10, IL-13) and Th3 (TGF-β). An increase in the production of proinflammatory cytokines and a decrease in the action of anti-inflammatory cytokines occurs in the pathogenesis of many neuro-inflammatory diseases such as Alzheimer’s disease [64,65], autoimmune encephalomyelitis/multiple sclerosis [66] and ischemic stroke [67,68].

### 3.1. Interleukin-1β

In our experiment, pre- and neonatal exposure of rats pups to Pb caused an increase in IL-1β concentration in the forebrain cortex, hippocampus and cerebellum. The results are consistent with literature data showing Pb-induced neuroinflammation in the hippocampus of rat pups [58] and mouse pups [69] resulting in an increase in IL-1β in this brain structure. Elevated IL-1β levels were also observed in the cerebral cortex of mouse pups [70]. This effect of Pb may be associated with the activation of microglial cells by Pb, which has been shown in vitro in murine microglial cells [61] and primary rat microglial cells [60], where IL-1β was also increased. However, there is no data available on the effect of Pb on IL-1β production in the remaining brain structures investigated in our study, i.e., the cerebellum and forebrain cortex.

It has been shown that IL-1β is important in the neurotoxicity of LPS [71,72] and secondary brain injury caused by neuroinflammation during traumatic brain injury [73] and ischemia [74,75,76]. Although IL-1β and neuroinflammation may act protectively in certain brain injuries [77], for example improving the function of the blood–brain barrier after injury [78,79]. Chronic and uncontrolled neuroinflammation has a destructive effect on brain tissue [53]. By analogy, a significant Pb-induced increase in IL-1β in rat pups suggests a significant neuroinflammatory effect of Pb. Chronic elevation of IL-1β can have severely negative effects in the developing brain. It causes axonal and dendritic damage [80] and myelin loss [48,81], as well as numerous detrimental effects at the cellular level, including apoptosis of oligodendrocyte progenitor cells [80] and oligodendrocytes [82], apoptosis of neurons [82] and reduced neural stem cell numbers [81]. This affects brain development, leading to learning and cognitive deficits in adult laboratory animals [81,83]. Similar effects have been shown in laboratory animals prenatally exposed to Pb [60,70].

### 3.2. IL-6

In this study chronic pre- and neonatal exposure of rats to Pb caused an increase in the production of IL-6 in the forebrain cortex, hippocampus and cerebellum, which is somewhat consistent with literature data. For example, it has been shown that maternal Pb exposure increases the expression of IL-6 in the murine hippocampus [69], confirmed by a study on BV-2 murine microglial cells [61]. Another study showed an increase in IL-6 mRNA throughout the whole brains of mouse pups exposed to very high doses of Pb, however IL-6 protein decreased in the hippocampus and did not change in the cerebellum and frontal cortex [84]. In contrast, Sobin et al. [85] and Struzynska et al. [60] showed that exposure to Pb either decreased the production of IL-6 in the posterior brain and hippocampus of young mice or did not affect the level of IL-6 in the hippocampus of young rats. However, these results were obtained from experiments in which animals were given Pb only in the neonatal period, i.e., after birth. In our work we treated rats from the first day of gestation, which suggests that prenatal exposure may be critical for the induction of neuroinflammation by Pb.

Due to the varied action of IL-6, its Pb-induced increase in different parts of the brain can have both positive and negative consequences. IL-6 has been shown to exert both a cytoprotective [71,85,86] and destructive pro-inflammatory [87,88] effect in various studies of neuroinflammation induced by LPS. It is likely that in our research model IL-6 regulates excessive neuroinflammation and protects against extensive damage to brain tissue, which requires clarification through further research on the effects of IL-6 on the brain of animals treated with Pb.

### 3.3. TGF-β

Our research showed for the first time that chronic exposure of rat pups to low Pb concentrations in the fetal and neonatal period caused an increase in TGF-β production in the forebrain cortex, hippocampus and cerebellum. These results are partially consistent with a report where exposure of mice to high doses of Pb in the fetal and postnatal period increased the amount of TGF-β1 in the frontal cortex and cerebellum, although not in the hippocampus [84]. The discrepancy in the results (i.e., concerning the hippocampus) may be related to differences in the experimental model, as Kasten-Jolly et al. 2011 examined mice and not rats, and PbAc was administered in drinking water at a concentration of 0.1 mM, much higher than the 0.1% PbAc used in our study. Therefore, the concentration of Pb in the blood of examined mice must have been much higher than in our experiment, resulting in a greater degree of neuroinflammation which then caused no change in the level of anti-inflammatory TGF-β in the hippocampus.

In our experiment, an increase in the level of TGF-β in the examined rat brain structures could have a neuroprotective effect and serve as a mechanism to offset the chronic toxic action of neuroinflammation in brain cells. TGF-β is a highly anti-inflammatory cytokine [89,90,91] and protects brain cells against factors and diseases such as ischemia [92], β-amyloid (Aβ) [93,94], hypoxia [95], LPS-induced neuroinflammation [96] and multiple sclerosis [97,98,99]. TGF-β also supports nerve regeneration [100]. An increase in the production of this cytokine occurs in the course of many brain diseases. For example, on the 7th day after ischemia there is an increase in TGF-β1 production, which controls disease progression and reduces the negative effects of the associated neuroinflammation [101,102]. A similar mechanism occurs in Alzheimer’s disease [93,94], when the increased production of TGF-β protects the brain from excessively rapid progression of this neurodegenerative disease.

The Pb-induced increase in the production of TGF-β also has negative effects during brain development. TGF-β is a tissue hormone that not only soothes inflammation and has a cytoprotective effect on differentiated brain cells, but is also an important factor in brain tissue development [103]. TGF-β is particularly important in the differentiation of neural stem cells [104] and radial glia [105]. For this reason, an increase in the production of TGF-β caused by Pb during brain development (in the fetal period) may lead to cerebral tissue malformations. This is confirmed by studies on the association between prenatal exposure to Pb and further cognitive development and some neurological diseases in children [19,20].

### 3.4. NF-κB

Our results showed for the first time that pre- and neonatal exposure to Pb increased nuclear factor κB (NF-κB) expression in the forebrain cortex, hippocampus and cerebellum. Previous research in vitro showed a Pb-induced reduction in the concentration of NF-κB and the inhibitor of NF-κB kinase (IKK) in primary fetal rat nerve cells [106]. This effect of Pb has never before been studied in in vivo experiments, so it is not possible to refer to any literature data, and so we can only infer that Pb likely increased the number of cells with the elevated expression of NF-κB. It might have also caused more frequent apoptosis of cells with lower expression of NF-κB. NF-κB is a transcription factor which not only plays a role in inflammatory reactions but also in cell survival related to its ability to directly increase the expression of Bcl-xL and Bfl-1/A1, antiapoptotic Bcl-2 proteins [107]. It is also possible that Pb-induced neuroinflammation results in the production of chemokines which cause activation of various cells in the brain that show high expression of NF-κB. However, these hypotheses still need to be verified by further studies.

### 3.5. Prostanoids

The results of our experiment showed that pre- and neonatal exposure to Pb increased the mRNA and protein expression of COX-1 and COX-2 in the forebrain cortex, hippocampus and cerebellum. There was also an increase in the products of these enzymes, i.e., PGE_2_ and TXB_2_. There is a lack of research on the effects of Pb on the expression of COX-1 or COX-2 in the brain. We only know that Pb caused neuroinflammation in BV-2 murine microglial cells in vitro, which was accompanied by an increase in the expression of COX-2 protein [61]. These observations are consistent with the results of our in vivo experiment.

The production pathway of the studied prostanoids (PGE_2_ and TXB_2_) consists of two stages. First, PGH_2_ is produced from arachidonic acid by COX-1 and COX-2; then a prostanoid is synthesized by an appropriate synthase [108]. COX-1 and COX-2 catalyze the same reaction. Nevertheless, they are associated with individual prostanoid synthases in different proportions, and so they are responsible for the synthesis of each prostanoid to a different degree. It was shown that cerebral overexpression of COX-1 causes a large increase in the production of PGI_2_ and TXB_2_, but only a small increase in the production of PGE_2_ [109]. In a *PTGS1* gene knockout, the production of PGE_2_ in the brain increased by 37% and the production of TXB_2_ decreased by 65% [110]. Therefore, in the brain COX-1 is responsible for the production of most of TXB_2_ and to a lesser extent for the production of PGE_2_. On the other hand, overexpression of COX-2 in the brain causes an increase in PGE_2_ and PGF_2α_ production [111,112]. So COX-2 is mainly responsible for the production of PGE_2_ and PGF_2α_ in the brain and partly for the production of TXB_2_. Therefore an increase in COX-1 and COX-2 expression will be associated with an increase in the amount of both PGE_2_ and TXB_2_, which is confirmed by our results.

Increased expression of COX-1 and COX-2 has a significant influence on brain tissue, resulting in elevated production of various prostanoids, which vary in their effects on brain tissue. Some of them, for example prostaglandin D_2_ (PGD_2_), have neuroprotective effects [113], while some, such as the active form of TXB_2_, support neuroinflammation, in particular by increasing IL-1β production, inducible nitric oxide synthase (iNOS) expression, and nitric oxide (NO) production in microglial cells exposed to Pb [114]. The action of the active form of TXB_2_ is also crucial in inflammation caused by ischemia [115] and LPS [116], and impairs the integrity of the blood–brain barrier in neuroinflammatory diseases [117].

PGE_2_ may have both positive and negative effects, depending on the physiological context. Under normal conditions, PGE_2_ exerts a cytoprotective effect through its EP2 receptor [118]. It is produced in the brain because it regulates membrane excitability and thus participates in hippocampal long-term synaptic plasticity [119,120]. However, in neuroinflammation, IL-1β increases the expression of another receptor, EP3 [121], which releases neurotoxic glutamate in response to PGE_2_ [122,123]. A similar toxicity mechanism occurs when the EP1 receptor is activated by PGE_2_ [124]. For this reason, increased activity of COX-2 and increased production of PGE_2_ during neuroinflammation causes apoptosis of cells in the brain. In the developing brain, increased PGE_2_ levels during neuroinflammation have been shown to have a clearly negative effect. Elevated PGE_2_ causes apoptosis of neural stem cells leading to developmental defects in the dentate gyrus and dysfunction of the hippocampus [52]. Increased concentration of PGE_2_ also inhibits maturation of oligodendrocyte progenitor cells, further disrupting brain development [46]. However, under physiological conditions, PGE_2_ is a signaling molecule that plays an important role in brain development, regulating proliferation and differentiation of neuroectodermal stem cells by affecting the Wnt/β-catenin pathway and is thus significant for the expression of genes important for brain development [57]. It is postulated that this is the mechanism behind the role of PGE_2_ in neurodevelopmental diseases, for example in ASD [57], which could explain the increased risk of this disorder in children with elevated blood Pb levels [23,24,25,26].

## 4. Materials and Methods

### 4.1. Reagents

The following antibodies were used in the current study: primary antibodies direct against COX-1 and COX-2 (1:200; Santa Cruz Biotechnology, Michigan, USA) or with a monoclonal anti-β-actin (1:200; Sigma Aldrich, Poznan, Poland) and next with a secondary antibody (goat anti-mouse IgG HRP, 1:2000; Santa Cruz Biotechnology, Michigan, USA). The Assay E2 ELISA Kit–Monoclonal (colorimetric) (Cat No. 514010) and Thromboxane B2 ELISA Kit (colorimetric) (cat. No. 501020) were purchased from Cayman, Michigan, USA. The ELISA Assay Kits IL-1b (cat.no. ELR-IL1b-CL1), IL-6 (cat no. ELR-IL6-CL1) were purchased from Raybiotech, Poznan, Poland. The ELISA Assay Kits NF-Kappa-B (Cat. No. LS-F12148-1) and TGF Beta 1 (Cat. No. LS-F24973-1) were obtained from LifeSpan BioSciences, Poznan, Poland. The RNeasy Lipid Tissue Mini Kit was obtained from Qiagen (Poznań, Poland). Reagents for reverse transcription (FirstStrand cDNA synthesis kit with oligo-dT primers) and PCR (Power SYBR Green PCR Master Mix) were obtained from Fermentas and Applied Biosystems (Foster City, CA, USA).

### 4.2. Animals

Animal procedures were performed in strict accordance with international standard guidelines for the care of animals and every effort was made to minimize suffering and the number of animals used. The experiments were approved by the Local Ethical Committee for Animal Research at the Pomeranian Medical University in Szczecin (Approval No. 5/2014, approval data 23 April 2014). Three-month-old female (250 ± 20 g) Wistar rats (*n* = 6) were kept for one week in a cage with sexually mature males (2:1). All animals had free access to food and water and were kept in a temperature-controlled enclosure according to a LD 12/12 regime. After one week the males were separated and each female was placed in a separate cage. Pregnant females were divided into two groups: control and experimental. Females from the experimental group (*n* = 3) received 0.1% lead acetate (PbAc) in drinking water ad libitum from the first day of pregnancy. The PbAc solution was prepared daily in disposable plastic bags (hydropac, Anilab, Poland) directly from solid reagent of the desired concentration and was not acidified. Pregnant females from the control group (*n* = 3) received distilled water until weaning. Liquid intake did not differ significantly between the experimental and control groups. The offspring (males and females) remained with and were breastfed by their mothers. During feeding the young mothers from the experimental group continued receiving PbAc in drinking water ad libitum. Young rats were weaned on the 21st post-natal day (PND 21) and placed in separate cages. From then on, young rats from both the experimental and control groups received only distilled water ad libitum until 28th post-natal day (PND 28).

The method of exposure (0.1% PbAc in drinking water) was selected as it mimics environmental exposure and is commonly used as a model for lead poisoning in animals [125,126]. In addition, our previous studies [8,9] revealed that this exposure protocol causes Pb concentration in whole blood (Pb-B) of rat offspring below the “threshold level” of 10 µg/dL [29]. Since the aim of the present study was to reach a Pb-B level below this threshold, we stopped administering Pb-B after weaning, and the rat pups were anesthetized (28 PND) and tissues were collected for Pb in whole blood (Pb-B) and molecular examination.

We randomly selected 18 young animals from control group and 18 from study group for Western blot, RT-PCR, ELISA method). No significant differences were found between the young females (*n* = 18) and males (*n* = 18) in terms of measured parameters (*p* = 0.5, Fisher exact test); therefore, pups were used regardless of gender. The percentage of males and females in the experimental and control groups did not differ significantly (*p* = 0.5, Fisher exact test). The weight of males (70–106 g) and females (52–85.5 g) in the experimental and control groups did not differ significantly (*p* = 0.5, Fisher exact test).

The animals were sacrificed without anesthesia by decapitation with scissors; the brains were quickly removed and divided into three sections: cerebellum (C), hippocampus (H) and forebrain cortex (FC), and then placed in liquid nitrogen. Samples were stored at −80 °C for further analysis.

### 4.3. Atomic Absorption Spectroscopy Pb Determination

Tissue lead content was analyzed by graphite furnace atomic absorption spectrometry (GFAAS) using a Perkin Elmer 4100 ZL spectrometer (Perkin Elmer, Warsaw, Poland) with Zeeman correction. Brain samples were digested at 120 °C for 16 h in a closed Teflon container with 1 mL of 65% HNO_3_. After cooling the sample, 1 mL of 30% H_2_O_2_ was treated and digested for the next 24 h under the same conditions.

The resulting solution was diluted to 10 mL with deionized water and analyzed by GFAAS together with the blank and control samples. Whole blood samples were deproteinized with 65% HNO_3_ and further analyzed as described above. The detection limit was 0.2 μg/dL.

### 4.4. Western Blotting Analysis of COX-1 and COX-2 Expression

RIPA buffer (pH 7.4) containing: 20 mM Tris; 0.25 mM NaCl; 11 mM EDTA; 0.5% NP-40, 50 mM sodium fluoride and protease, phosphatase inhibitors (Sigma, Poznań, Poland) was used to homogenize brain samples [127]. Total protein concentrations were determined using the MicroBCA Protein Assay Kit, the homogenates underwent electrophoresis in SDS-polyacrylamide gel and COX-1 and COX-2 protein expression were examined. In short: extracted proteins (20 µg/well) were separated into 12% gel (SDS-PAGE) using Mini Protean Tetra Cell system (Bio-Rad, Poznań, Poland). Fractionated proteins were transferred to a 0.2 µm polyvinylidene difluoride (PVDF) membrane (Bio-Rad, Poznań, Poland). The membranes were then blocked with 3% bovine serum albumin (BSA) in buffer for 1 h at room temperature. Protein expression in brain tissue was determined by immunodetection with specific antibodies. The membranes were developed using the ECL Advance Western Blotting Detection Kit (Amersham Life Sciences, Pittsburgh, PA, USA) and then bands were visualized with the Gel DOC-It imaging system.

### 4.5. Quantitative Real-Time PCR Analysis (qRT-PCR) of COX-1 and COX-2 mRNA

Quantitative analyses of COX-1 and COX-2 mRNA expression were performed by two-stage PCR with reverse transcription. Total RNA was extracted from 50–100 mg of tissue sample using the RNeasy Lipid Tissue Mini Kit (Qiagen, Poznań, Poland). cDNA was prepared from 1 μg total cellular RNA in 20 μL reaction volume using the FirstStrand cDNA synthesis kit and oligo-dT primers (Fermentas, Waltham, Massachusetts, USA). Quantitative evaluation of mRNA levels was performed by RT-PCR in real time, using ABI 7500Fast with Power SYBR Green PCR Master Mix reagent. Real-time conditions were as follows: 95 °C (15 s), 40 cycles at 95 °C (15 s) and 60 °C (1 min). According to melting point analysis, only one PCR product was amplified under these conditions. Each sample was analyzed in two technical replicates and the mean Ct values were used for further analysis. The relative target quantity, standardized to the endogenous glyceraldehyde-3-phosphate dehydrogenase (*GAPDH*) control gene and to the calibrator, was expressed as 2^−∆∆*C*t^ (-fold difference), where Ct is the threshold cycle, ∆Ct = (target gene Ct) − (endogenous control gene Ct) and ∆∆Ct = (∆Ct target gene sample) − (∆Ct target gene calibrator). The following pairs of primers were used- for *PTGS1* (COX-1 gene): 5’-GTTCACAGAGGAGAGAGAGAGAGATG-3’ and 5’-GGAGCCCCCCCCCATCTCTCTATCATCATGC-3’; For *PTGS2* (COX-2 gene)): 5’-AATGAGTACCGCAAACGCTTCTCTCT-3’ and 5’-AGCCATTTCTTCTCTCTCTTGTAAG-3’; for *GAPDH*: 5’-AGG ATC GCA ATG TGG CCA CTC-3’ and 5’- TCC TTTT TCG TAG TAA TGC TGC TGC -3’.

### 4.6. Measurements of Prostaglandin (PGE_2_) and Thromboxane B2 (TXB_2_) Concentrations by ELISA Method

Activity of COX-1 and COX-2 was tested in vitro by quantitative measurement of their products: PGE_2_ and TXB_2_. Prostanoids were extracted from tissues using Bakerbond SPE columns (J.T. Baker, USA) as described in the manufacturer’s instructions. PGE_2_ and TXB_2_ concentrations were determined by modified ELISA immunoassay using the Prostaglandin E_2_ EIA Kit-Monoclonal Cayman Kit (USA) according to the manufacturer’s instructions. Since thromboxane A_2_ (TXA_2_) has a short half-life (37 s) and is rapidly hydrolyzed non-enzymatically to its stable TXB_2_ derivative, the TXB_2_ enzyme immunoassay kit (Thromboxane B_2_ EIA Kit, Cayman, Ann Arbor, USA), used according to the protocol provided by the manufacturer) was used to measure free TXA_2_ indirectly.

### 4.7. Measurement of Interleukin 1β (IL-1β) and Interleukin 6 (IL-6) Concentration by ELISA Method

The measurements of IL-1β and IL-6 concentrations were conducted using appropriate immunoenzymatic ELISA Assay Kits IL-1beta (cat.no. ELR-IL1b-CL1) and IL-6 (cat no. ELR-IL6-CL1), (Raybiotech, Poznan, Poland) according to manufacturer’s instruction.

### 4.8. Measurement of Transforming Growth Factor-beta 1 (TGF-β1) and nuclear factor kappa B (NF-κB) concentration by ELISA method

The measurement of TGF-β1 and NF-κB concentration were conducted using appropriate immunoenzymatic ELISA Assay Kits NF-Kappa-B (Cat. No. LS-F12148-1) and TGF Beta 1 (Cat. No. LS-F24973-1), (LifeSpan BioSciences, Poznan, Poland) according to manufacturer’s instruction.

### 4.9. Protein Assay

The expression of COX-1 and COX-2 protein and the concentration of TXB_2_, PGE_2_, IL-1, IL-6, NF-κB and TGF-β were calculated from protein content in the samples measured using the MicroBCA protein kit (Thermo Scientific, Pierce Biotechnology, Waltham, Massachusetts, USA) and plate reader (UVM340, ASYS). This assay kit is a two-component, precision, detergent-compatible set of reagents used to measure total protein concentration compared to the protein standard. This method combines the well-known reduction of Cu^2+^ to Cu^1+^ by protein in an alkaline environment with highly sensitive and selective colorimetric detection of copper cation (Cu^1+^) using bicinchoninic acid (BCA) [128].

### 4.10. Statistical Analysis

The obtained results were analyzed using the Statistica 10.0 software package. The arithmetical mean ± SD was calculated for each of the studied parameters. The distribution of results for individual variables was obtained with the Shapiro–Wilk W test. As most of the distributions deviated from the normal distribution, nonparametric tests were used for further analyses. To assess the differences between the groups studied, the nonparametric Mann–Whitney U-test was used. The Spearman correlation rank coefficient was used to determine the strength of correlations between the parameters. A probability at *p* ≤ 0.05 was considered as statistically significant.

## 5. Conclusions

Pre- and neonatal exposure of rats to lead results in increased levels of proinflammatory cytokines (IL-1β and IL-6), increased expression of COX-1 and COX-2 and increased production of prostanoids (PGE_2_ and TXB_2_), as well as an increased concentration of NF-κB in the forebrain cortex, hippocampus and cerebellum.

The increase in TGF-β concentration in the studied brain structures may have anti-inflammatory and neuroprotective effects.

## Figures and Tables

**Figure 1 ijms-21-01083-f001:**
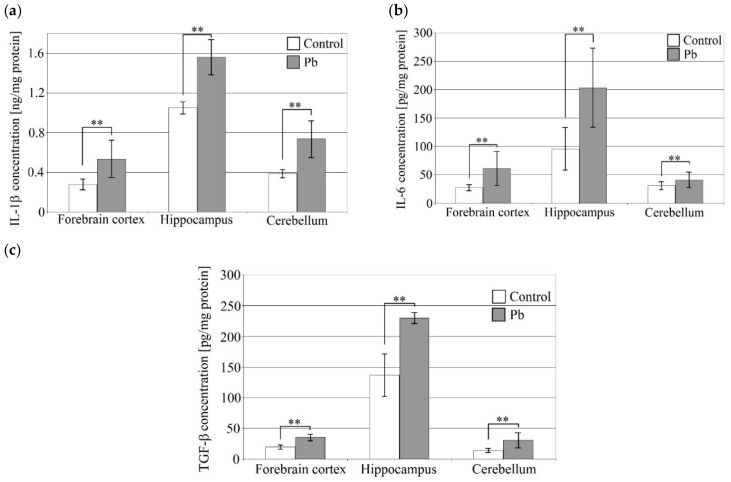
The effect of perinatal exposure to Pb on cytokine concentrations in the rat brain. Concentrations of IL-1β (**a**), IL-6 (**b**), TGF-β (**c**) in the forebrain cortex, hippocampus and cerebellum. From the first day of pregnancy, rats were given distilled water (control) or 0.1% PbAc in drinking water ad libitum. After birth, the Pb compound was continued until 21th post-natal day (PND 21). Young rats were weaned on the PND 21 and placed in separate cages. From then on, young rats from both the experimental and control groups received only distilled water ad libitum until 28 PND. After this time, the rat pups were anesthetized and tissues were collected for examination. Following tissue homogenization, selected cytokine concentrations were analyzed by ELISA. Data represent the means ± standard deviation (SD), *n* = 6 animals per parameter measured (total *n* = 18). ** *p* < 0.005 versus control using Mann–Whitney U-test.

**Figure 2 ijms-21-01083-f002:**
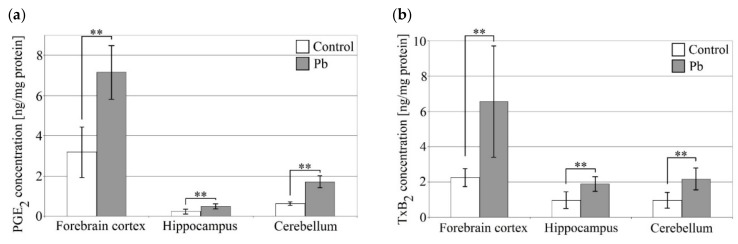
The effect of perinatal exposure to Pb on prostanoid concentrations in the rat brain. Concentration of prostaglandin E2 (PGE2) (**a**) and thromboxane B2 (TXB2) (**b**) in the forebrain cortex, hippocampus and cerebellum. From the first day of pregnancy, rats were given distilled water (control) or 0.1% PbAc in drinking water ad libitum. After birth, the Pb compound was continued until 21th post-natal day (PND 21). Young rats were weaned on the PND 21 and placed in separate cages. From then on, young rats from both the experimental and control groups received only distilled water ad libitum until 28 PND. After this time, the rat pups were anesthetized and tissues were collected for examination. Following tissue homogenization, selected cytokine concentrations were analyzed by ELISA. Data represent the means ± SD, *n* = 6 animals per parameter measured (total *n* = 12).** *p* < 0.005 versus control using Mann–Whitney U-test.

**Figure 3 ijms-21-01083-f003:**
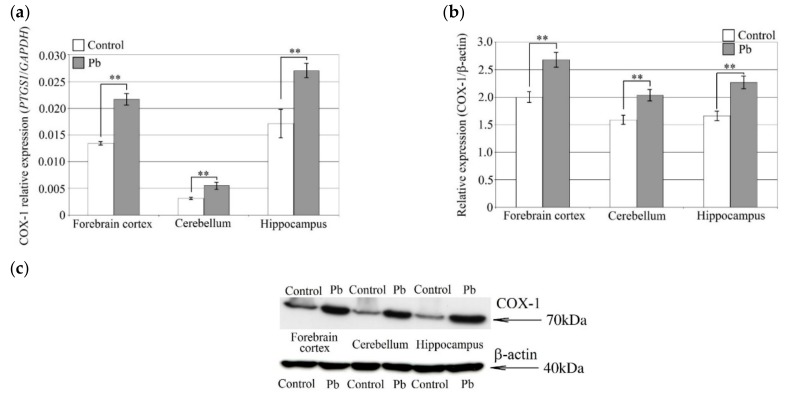
The effect of perinatal exposure to Pb on COX-1 expression in the rat brain. COX-1 mRNA expression (**a**) and COX-1 protein expression (**b**) and representative Western blot results (**c**) for forebrain cortex, hippocampus and cerebellum in rat pups treated with PbAc. From the first day of pregnancy, rats were given distilled water (control) or 0.1% PbAc in drinking water ad libitum. After birth, the Pb compound was continued until 21th post-natal day (PND 21). Young rats were weaned on the PND 21 and placed in separate cages. From then on, young rats from both the experimental and control groups received only distilled water ad libitum until 28 PND. After this time, the rat pups were anesthetized and tissues were collected for examination. Following tissue homogenization COX-1 mRNA and protein expression was analyzed by qRT-PCR and Western blot, respectively. Data represent the means ± SD, *n* = 6 animals per parameter measured (total *n* = 12). ** *p* < 0.005 versus control using a Mann–Whitney U-test.

**Figure 4 ijms-21-01083-f004:**
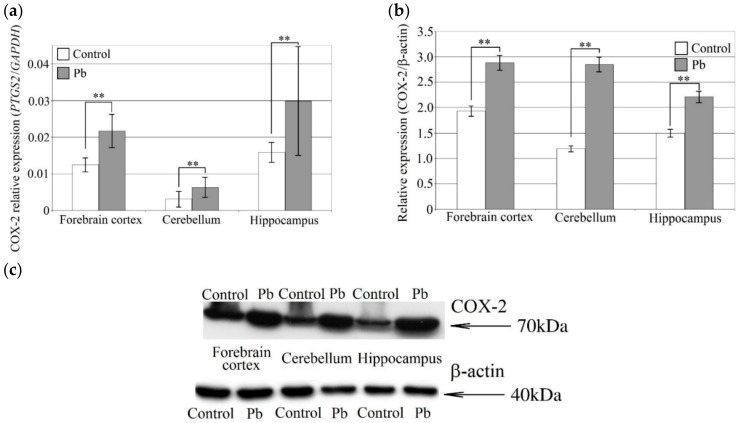
The effect of perinatal exposure to Pb on COX-2 expression in the rat brain. COX-2 mRNA (**a**) and COX-2 protein expression (**b**) and representative Western blot result (**c**) for forebrain cortex, hippocampus and cerebellum in rat pups treated with PbAc. From the first day of pregnancy, rats were given distilled water (control) or 0.1% PbAc in drinking water ad libitum. After birth, the Pb compound was continued until 21th post-natal day (PND 21). Young rats were weaned on the PND 21 and placed in separate cages. From then on, young rats from both the experimental and control groups received only distilled water ad libitum until 28 PND. After this time, the rat pups were anesthetized and tissues were collected for examination. Following tissue homogenization COX-2 mRNA and protein expression were analyzed by qRT-PCR and Western blot, respectively. Data represent the means ± SD, *n* = 6 animals per parameter measured (total *n* = 12). ** *p* < 0.005 *versus* control using a Mann–Whitney U-test.

**Figure 5 ijms-21-01083-f005:**
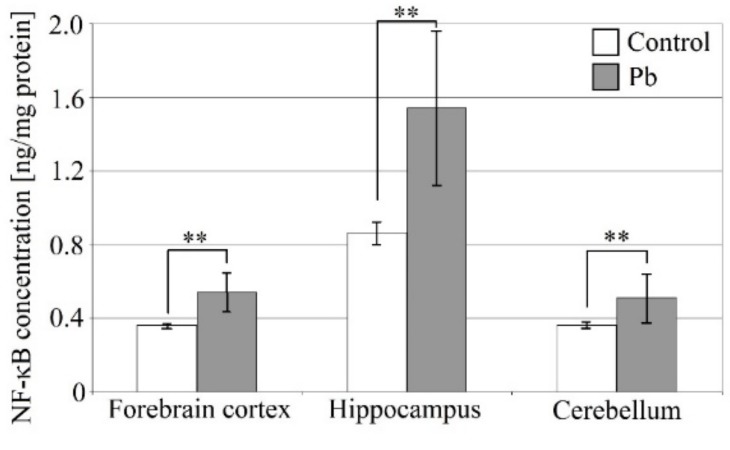
The effect of perinatal exposure to Pb on the level of NF-κB in the rat brain. From the first day of pregnancy, rats were given distilled water (control) or 0.1% PbAc in drinking water ad libitum. After birth, the Pb compound was continued until 21th post-natal day (PND 21). Young rats were weaned on the PND 21 and placed in separate cages. From then on, young rats from both the experimental and control groups received only distilled water ad libitum until 28 PND. After this time, the rat pups were anesthetized and tissues were collected for examination. Following tissue homogenization NF-κB analysis was performed by ELISA. Data represent the means ± SD, *n* = 6 animals per parameter measured. ** *p* < 0.005 versus control using a Mann–Whitney U-test.

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
