# Peer review of "Pre- and Neonatal Exposure to Lead (Pb) Induces Neuroinflammation in the Forebrain Cortex, Hippocampus and Cerebellum of Rat Pups"

_ijms, 2020, doi:10.3390/ijms21031083_

Round 1

Reviewer 1 Report

The Authors have adequately addressed my criticisms

Author Response

We are very grateful for the review

Reviewer 2 Report

This is a straightforward paper that describes the effects of Pb exposure in rat pups (in utero and neonatal exposure). There is already ample evidence that Pb causes brain inflammation, but some of the brain regions examined here have not be specifically reported before, so while somewhat incremental in nature, the study does add some new information to this area. There are some issues with the description of methods, and with data presentation that need to be fixed.

Specific comments:
It is not clear when the blood lead levels were determined. Was blood sampled at birth, before weaning or at sacrifice on PND28? This is important information because offspring blood lead levels could be substantially different in utero vs. breast feeding, vs. one week of drinking normal water. This should be more clearly described.

The exposure paradigm is not clearly described. In the results section it is stated that the weaned pups continue to receive Pb in drinking water until they are sacrificed: “After birth, the Pb compound was continued until 28th post-natal day (PND 28)”. In the methods section it is stated that weaned pups are maintained on Pb-free drinking water until they are sacrificed: “From then on, young rats from both the experimental and control groups received only distilled water ad libitum until 28th post-natal day (PND 28).”. This are dramatically different circumstances that have a significant impact on how the results should be interpreted. Were different exposure paradigms used in different experiments, or were all animals exposed according to the same agenda.

Greater detail should be provided about the antibodies used to detect COX1 and COX2. What are the catalog/item numbers and how was specificity evaluated. A related issue is that the western blot panels shown in Fig. 3 and Fig. 4 should include the locations of size markers.

Author Response

Reviewer 2

This is a straightforward paper that describes the effects of Pb exposure in rat pups (in utero and neonatal exposure). There is already ample evidence that Pb causes brain inflammation, but some of the brain regions examined here have not be specifically reported before, so while somewhat incremental in nature, the study does add some new information to this area. There are some issues with the description of methods, and with data presentation that need to be fixed.

Specific comments:
It is not clear when the blood lead levels were determined. Was blood sampled at birth, before weaning or at sacrifice on PND28? This is important information because offspring blood lead levels could be substantially different in utero vs. breast feeding, vs. one week of drinking normal water. This should be more clearly described.

The exposure paradigm is not clearly described. In the results section it is stated that the weaned pups continue to receive Pb in drinking water until they are sacrificed: “After birth, the Pb compound was continued until 28th post-natal day (PND 28)”. In the methods section it is stated that weaned pups are maintained on Pb-free drinking water until they are sacrificed: “From then on, young rats from both the experimental and control groups received only distilled water ad libitum until 28th post-natal day (PND 28).”. This are dramatically different circumstances that have a significant impact on how the results should be interpreted. Were different exposure paradigms used in different experiments, or were all animals exposed according to the same agenda.

We are very grateful for this remark, according to Reviewer we corrected the materials and methods section and Figures captions. We added also graphical abstract with Pb exposure regimen used in our study.

“Young rats were weaned on the 21st post-natal day (PND 21) and placed in separate cages. From then on, young rats from both the experimental and control groups received only distilled water ad libitum until 28th post-natal day (PND 28). The method of exposure (0.1% PbAc in drinking water) was selected as it mimics environmental exposure and is commonly used as a model for lead poisoning in animals [125,126]. In addition, our previous studies [8,9] revealed that this exposure protocol causes Pb concentration in whole blood (Pb-B) of rat offspring below the "threshold level" of 10 µg/dL [29]. Since the aim of the present study was to reach a Pb-B level below this threshold, we stopped administering Pb-B after weaning, and the rat pups were anesthetized (28 PND) and tissues were collected for Pb in whole blood (Pb-B) and molecular examination.”

Greater detail should be provided about the antibodies used to detect COX1 and COX2. What are the catalog/item numbers and how was specificity evaluated. A related issue is that the western blot panels shown in Fig. 3 and Fig. 4 should include the locations of size markers.

 According to Reviewer remark we added cat no of antibodies used in our study and supplemented figures with size markers.

"The following antibodies were used in the current study: primary antibodies direct against COX-1 (cat. no sc-19998) and COX-2 (cat. no sc-19999); (in dilution 1:200; Santa Cruz Biotechnology, Michigan, USA) or with a monoclonal anti-β-actin (cat no A3854), (in dilution 1:200; Sigma Aldrich, Poznan, Poland) and next with a secondary antibody (goat anti-mouse IgG HRP, cat no sc-2005); (in dilution 1:2000; Santa Cruz Biotechnology, Michigan, USA)”.
